# Absolute Configuration and Biological Evaluation of Novel Triterpenes as Possible Anti-Inflammatory or Anti-Tumor Agents

**DOI:** 10.3390/molecules27196641

**Published:** 2022-10-06

**Authors:** Zhenzhen Wei, Tiqiang Zhou, Ziming Xia, Sifan Liu, Min Li, Guangjie Zhang, Ying Tian, Bin Li, Lin Wang, Shuchen Liu

**Affiliations:** 1Faculty of Environment and Life, Beijing University of Technology, Beijing 100022, China; 2Department of Pharmaceutical Science, Beijing Institute of Radiation Medicine, Beijing 100850, China; 3School of Life Science, Beijing Institute of Technology, Beijing 100081, China

**Keywords:** *Ardisia lindleyana* D. Dietr, triterpene, ardisiapunine B, ardisiapunine C, absolute configurations, single–crystal X–ray diffraction, biological activity

## Abstract

Two new compounds, ardisiapunine B (**1**) and ardisiapunine C (**2**), were isolated from *Ardisia lindleyana* D. Dietr. Their structures were examined using HR–ESI–MS, IR, (1D, 2D) NMR spectroscopic analyses, single–crystal X–ray diffraction, and ECD calculation. It was found that the two new compounds belong to unusual oleanane-type triterpenes, with compound **1** bearing an acetal unit and a C–13–C–18 double bond, and compound **2** bearing a C–28 aldehyde group and a C–18–C–19 double bond. The anti-inflammatory properties of compounds **1** and **2** were tested on NO production and cellular morphology using RAW264.7 cells, and their anti-tumor properties were tested on cytotoxic activities, cellular morphology, cell apoptosis, and cell cycle. The results showed that compound **1** exhibited a potent cytotoxicity against HepG2 cell lines with an IC_50_ of 12.40 μM. Furthermore, it is possible that compound **1** inhibits cell proliferation by blocking the cell G2/M phase and promoting cell apoptosis. Compound **2** exhibited a potential anti-inflammatory activity by decreasing the production of NO in LPS–stimulated RAW264.7 cells. Comparative analysis of the structures of compounds **1** and **2** revealed that the acetal structure and double bond positions were the main differences between them, and these are presumed to be the main reasons for the extreme differences in their cytotoxicity and anti-inflammatory activities. From these new findings, two promising lead compounds were identified for the future development of potential anti–inflammatory or anti–tumor agents.

## 1. Introduction

*Ardisia lindleyana* D. Dietr, a kind of evergreen dwarf shrub, belongs to the family of Myrsinaceae. It is mainly distributed across tropical America, the Pacific Islands, the eastern Indian Peninsula, Eastern Asia, and Southern Asia [1], and is grown for use in traditional Miao Minority medicine and as an ornamental shrub [2]. Previous research on the chemical constituents of this plant has led to the isolation of isocoumarins [3], quinones [4,5], and triterpenoids [3], some of which were found to have various bioactivities, including anti–inflammatory and anti–tumor activities.

In preliminary studies, we reported that the ethanol extract of the plant had a significant inhibitory effect on the proliferation of multiple tumor cells in vitro, disclosing that this plant may contain natural products with anti–tumor properties [6]. In this paper, we report the results of our further chemical investigation of *Ardisia lindleyana* D. Dietr, which led to the isolation of two new oleanane–type triterpenes, one with a diacetal moiety (**1**) and another with an aldehyde group (**2**). Their structures, including absolute configurations, were characterized by one- and two-dimensional NMR spectroscopic analyses, including DEPT, HSQC, HMBC, ^1^H–^1^H COSY, and NOESY experiments, single–crystal X–ray diffraction, and ECD calculation. Details of the structural elucidations for compounds **1** and **2** are shown in Figure 1A. The structures of the new compounds **1** and **2**, both oleanane-type triterpenes, are novel, with compound **1** bearing an acetal unit and a C–13–C–18 double bond, and compound **2** bearing a C–28 aldehyde group and a C–18–C–19 double bond. The characterization of compounds **1** and **2** add not only to the diversity of the chemical constituents of *Ardisia lindleyana* D. Dietr, but also to the diversity of oleanane-type triterpenes.

Triterpenes are common in nature and engage in various biological activities, such as anti–tumor activity, anti–inflammatory activity, antibacterial activity, and antiviral activity. Among these, the anti-tumor activity of triterpenes is measured by their ability to induce apoptosis, block nuclear factor–kappaB activation, inhibit signal transducer, inhibit topoisomerases, and activate transcription and angiogenesis [7]. Hence, compounds **1** and **2** were evaluated for their cytotoxicity against B16F10, A549, H460, MCF–7, HepG2, Hela, and U87 cell lines in vitro, and evaluated for their anti-inflammatory properties using the Griess reaction and the results of screening the production of NO by LPS–stimulated murine macrophage cell lines RAW264.7.

## 2. Results and Discussion

### 2.1. Phytochemical Identification

Ardisiapunine B (**1**) was obtained as a white amorphous powder. The purity of compound **1** was more than 95% according to HPLC analysis. The IR spectrum showed absorption bands assignable to hydroxyl (3450 cm^−1^), double bond (1644 cm^−1^), alkyl (1460 cm^−1^), and methyl (1387 cm^−1^) functional groups. The HR-ESI-MS gave a molecular formula of C_31_H_50_O_4_Na at *m/z* 509.3598 [M + Na] ^+^ (calcd. for C_31_H_50_O_4_Na^+^, 509.3607) with 7 degrees of unsaturation. The ^13^C NMR spectrum of compound **1** showed 31 carbon signals (Table 1). Combined with the ^13^C NMR and DEPT (45, 90, 135) experiment, 31 carbon signals were categorized as follows: 2 sp^2^ olefinic carbons at *δ*_C_ 131.7 ppm and 134.1 ppm, 6 sp^3^ quaternary carbons at *δ*_C_ 38.9 ppm, 40.7 ppm, 40.9 ppm, 43.2 ppm, 45.7 ppm, and 45.8 ppm, 3 sp^3^ oxymethine carbons at *δ*_C_ 72.1 ppm, 79.3 ppm, and 109.7 ppm, 2 sp^3^ methine carbons at *δ*_C_ 52.8 ppm and 57.4 ppm, 1 oxymethylene carbon at *δ*_C_ 72.2 ppm, 10 sp^3^ methylene carbons at *δ*_C_ 20.2 ppm, 23.5 ppm, 25.9 ppm, 27.1 ppm, 29.6 ppm, 32.6 ppm, 36.1 ppm, 36.9 ppm, 38.1 ppm, and 40.8 ppm, 1 oxymethyl carbons at *δ*_C_ 56.2 ppm, and 6 methyl carbons at *δ*_C_ 17.7 ppm, 18.3 ppm, 20.2 ppm, 27.4 ppm, 29.9 ppm, and 30.0 ppm. The ^1^H NMR (Pyridine–*d5*, 400 MHz) spectrum (Table 1) of compound **1** displayed six methyl singlets at *δ*_H_ 0.88 ppm (3H, s, H–25), 0.98 ppm (3H, s, H–29), 1.00 ppm (3H, s, H–26), 1.02 ppm (3H, s, H–24), 1.20 ppm (3H, s, H–23), and 1.40 ppm (3H, s, H–27), a methoxy signal at *δ*_H_ 3.40 ppm (3H, s, H–31), an oxymethine signal at *δ*_H_ 3.47 ppm (1H, t, *J* = 7.5 Hz, H–3), an oxymethylene signal at *δ*_H_ 3.82 ppm (1H, t, *J* = 4.0 Hz, H–28), and a methinedioxy group at *δ*_H_ 4.40 ppm (1H, s, H–30). Four spin coupling systems (H_2_–2–H–3, H_2_–11–H_2_–12, H_2_–15–H–16, and H_2_–21–H_2_–22) were revealed by the analysis of the ^1^H–^1^H COSY data for compound **1** (Figure 1B). Combined with the ^1^H–^1^H COSY data, the HMBC correlations (Figure 1B) from H–3 to C–1/C–2/C–23/C–24, from H–12 to C–11/C–9/C–13/C–14, from H–5 to C–10/C–18, from H–19 to C–13/C–18/C–20/C–21, from H–30 to C–19/C–28/C–29/C–31, from H–28 to C–18/C–22, and from H–16 to C–17 revealed the partial structure of compound **1**. Based on the above observations, the molecular formula, the degree of unsaturation, and the chemical shift characteristics, the whole planar structure of compound **1** was assigned as shown in Figure 1A.

The relative stereochemistry of compound **1** was ascertained from the NOE interactions (Figure 1C) observed in the NOESY spectrum. Significant NOE correlations were observed between H–12 and H–19, H–19 and H–28, H–19 and H–30, H–28 and H–31, H–31 and H–30, and H–30 and H–21. These NOE interactions suggested that C–28, C–30, and C–31 were on the axial positions. The absolute configuration of compound **1** was further secured by a successful X–ray diffraction experiment (Flack parameter −0.004 (13)), which established the absolute configuration of compound **1** as 3*S*, 5*R*, 8*R*, 9*R*, 10*R*, 14*S*, 16*R*, 17*S*, 20*S*, and 30*S* (Figure 2). Thus, the structure of compound **1** was established and named ardisiapunine B.

Ardisiapunine C (**2**) was obtained as a white amorphous powder. The purity of compound **2** was more than 95% according to HPLC analysis. The IR spectrum showed absorption bands assignable to hydroxyl (3379 cm^−1^), methylene (2937 cm^−1^), aldehyde (2868 cm^−1^, 2723 cm^−1^ and 1715 cm^−1^), double bond (1650 cm^−1^), alkyl (1451 cm^−1^), and methyl (1384 cm^−1^) functional groups. The HR-ESI-MS gave a protonated molecule peak of *m/z* 495.3448 [M + Na] ^+^ (calcd. for C_30_H_48_O_4_Na^+^, 495.3450) and indicated the molecular formula of C_30_H_48_O_4_ (7 degrees of unsaturation). Thirty carbon signals were observed in the ^13^C NMR of compound **2** (Table 1). The DEPT spectra of compound **2** indicated the presence of two sp^2^ olefinic carbons at *δ*_C_ 132.1 ppm and 137.4 ppm, one sp^2^ aldehyde carbon at *δ*_C_ 205.9 ppm, two sp^3^ oxymethine carbons at *δ*_C_ 75.5 ppm and 77.8 ppm, one sp^3^ oxymethylene carbon at *δ*_C_ 71.5 ppm, six sp^3^ quaternary carbons, three sp^3^ methine carbons, nine sp^3^ methylene carbons, and six sp^3^ methyl carbons. The ^1^H NMR (Pyridine–*d5*, 400 MHz) spectrum of compound **2** (Table 1) showed six methyl singlets at *δ*_H_ 0.85 ppm (3H, s, H–25), 0.95 ppm (3H, s, H–27), 1.01 ppm (3H, s, H–26), 1.07 ppm (3H, s, H–24), 1.19 ppm (3H, s, H–23), and 1.24 ppm (3H, s, H–29), three oxymethylene signals at *δ*_H_ 3.42 ppm (m, H–3), 3.68 ppm (m, H–30), and 4.12 ppm (dd, *J* = 4.56, 11.92 Hz, H–16), and one double bond signal at *δ*_H_ 5.65 ppm (s, H–19). Extensive 2D NMR experiments allowed us to define the molecular connectivity. The analysis of the ^1^H–^1^H COSY experiment and the coupling constant values of the protons revealed the presence of four isolated spin systems, as shown in Figure 1B. Combined with the ^1^H–^1^H COSY data, the HMBC correlations (Figure 1B) from H–3 to C–4/C–23/C–24, from H–13 to C–12/C–8, from H-15 to C–8/C–16/C–17, from H-16 to C–17/C–22/C–28, from H–22 to C–13/C-17/C–28, and from H–19 to C–17/C–21/C–29/C–30 revealed the partial structure of compound **2**. Compound **2** is an oleanolane pentacyclic triterpene with ten chiral carbons. According to the basic chemical structure of 3–hydroxyl oleanane-type pentacyclic triterpenes (Figure 3C), the relative configurations of the nine chiral carbons in compound **2** were assigned as 3*S**, 5*R**, 8*R**, 9*R**, 10*R**, 13*S**, 14*R**, 17*R**, and 20*S**. The observed NOESY correlations (Figure 1C) between H–13 and H–28, and H–16 and H–28 indicated that H–13, H–16, and H–28 were on the same axial orientation. Therefore, the relative configuration of C–16 in compound **2** was 16*R**.

We had originally intended to verify the absolute configuration of compound **2** using the single–crystal diffraction method, but we failed to precipitate crystallization. Electronic circular dichroism (ECD) has become a popular method for assigning the absolute configuration of novel natural products [8,9]. However, the configuration of compound **2** cannot be determined directly using ECD due to weak UV absorption. Derivatization of carbonyl compounds with 2,4–dinitrophenylhydrazine (DNPH) is one of the most widely used analytical methods [10]. Therefore, compound **2** was converted into compound **2–DNP** using a diffraction method (Figure 3A), and because compound **2–DNP** had a chromophoric group for UV absorption, the absolute configuration of compound **2–DNP** was determined by ECD, thus verifying the absolute configuration of compound **2**.

The process of derivatization of compound **2** to compound **2–DNP** was as follows: A stirred solution of compound **2** (28 mg, 59 μmoL) was dissolved in a stirred solution of dichloromethane: methanol (1:1, *v*/*v*, 25 mL). Acetic acid (0.2 mL) and DNPH (18 mg, 91 μmoL) were added to the mixture at a temperature of 60 °C, then the mixture was stirred at the same temperature for 30 min. The residue was purified by Sephadex LH–20 column chromatography with methanol as the mobile phase to give compound **2–DNP** (23 mg, 35 μmoL). The HR–ESI–MS of compound **2–DNP** (Appendix A) gave a protonated molecule peak of *m/z* 651.3763 [M − H] ^−^ (calcd. for C_36_H_51_N_4_O_7_^−^, 651.3758) and indicated the molecular formula of C_36_H_52_N_4_O_7_ (13 degrees of unsaturation). The structure of compound **2–DNP** was confirmed using IR spectroscopy (Appendix A), which showed absorption bands at 1616 cm^−1^ (C=N), 1132 cm^−1^ (C–N), and 1330 cm^−1^ (–NO_2_). Additionally, the disappearance of a band at 1715 cm^−1^ was detected (C=O). The ^13^C NMR spectrum of compound **2–DNP** (Table 1) revealed the disappearance of carbon signals at 205.9 ppm (C=O), as well as the appearance of six peaks at *δ*_C_ 116.7 ppm, 122.7 ppm (overlapped), 129.0 ppm, 129.6 ppm, 137.2 ppm, and 145.6 ppm, and three signals at *δ*_H_ 8.12 ppm, 8.19 ppm, and 9.04 ppm in the ^1^H NMR spectra (Table 1), which were attributed to a benzene group. Moreover, the presence of hydrazone moiety (C=N-NH) was confirmed by the singlet peaks at 11.8 ppm and 157.4 ppm in the ^1^H NMR and ^13^C NMR spectra, respectively.

During the process of derivatization of compound **2** to compound **2–DNP**, the chiral carbons of compound **2** were not directly involved in this derivative reaction. The relative configurations of (3*S**, 5*R**, 8*R**, 9*R**, 10*R**, 13*S**, 14*R**, 16*R**, 17*S**, 20*S**)–2–DNP were determined by ECD calculation. The calculated ECD curve of (3*S*, 5*R*, 8*R*, 9*R*, 10*R*, 13*S*, 14*R*, 16*R*, 17*S*, 20*S*)–**2–DNP** was similar to that obtained in the experiment (Figure 3B). Thus, the absolute configuration of compound **2** was established as 3*S*, 5*R*, 8*R*, 9*R*, 10*R*, 13*S*, 14*R*, 16*R*, 17*R*, and 20*S*, and named ardisiapunine C.

In this work, two new oleanane–type triterpenes were obtained from *Ardisia lindleyana* D. Dietr. This is the first report about oleanane–type triterpenes with double bonds at C–13–C–18 or C–18–C–19 obtained from *Ardisia lindleyana* D. Dietr, which has broadened our understanding of the structural diversity of this genus of metabolites. To the best of our knowledge, although an oleanane–type triterpene containing an acetal moiety had been reported [11], there are no reports of one with the properties of compound **1**, bearing an acetal and a C–13–C–18 double bond, being obtained from natural sources. It is worth mentioning that such compounds, with a double bond at this position, are very special and rare. The structure of compound **2,** bearing a C–28 aldehyde group and a C–18–C–19 double bond, also appears to be unique.

### 2.2. Structural Information of Compounds **1** and **2**

Ardisiapunine B (**1**): White amorphous powder; UV (DMSO) *λ*_max_ 210 nm; HR–ESI–MS (positive) *m/z* 509.3598 [M + Na]^+^ (calcd. for C_31_H_50_O_4_Na^+^, 509.3607); IR (KBr) *ν*_max_ 3450 cm^−1^, 2942 cm^−1^, 1644 cm^−1^, 1460 cm^−1^, 1387 cm^−1^, 1044 cm^−1^, and 974 cm^−1^; for ^1^H and ^13^C NMR see Table 1.

Ardisiapunine C (**2**): White amorphous powder; UV (MeOH) *λ*_max_ 210 nm; HR–ESI–MS (positive) *m/z* 495.3448 [M + Na]^+^ (calcd. for C_30_H_48_O_4_Na^+^, 495.3450); IR (KBr) *ν*_max_ 3379 cm^−1^, 2937 cm^−1^, 2868 cm^−1^, 2723 cm^−1^, 1715 cm^−1^, 1650 cm^−1^, 1451 cm^−1^, 1384 cm^−1^, and 1036 cm^−1^; for ^1^H and ^13^C NMR see Table 1.

### 2.3. X-Ray Crystallographic Analysis of Compound **1**

C_31_H_50_O_4_ (M = 486.37 g/mol): The empirical formula containing one molecule of chloroform was C_32_H_51_C_l3_O_4_ (M = 606.08 g/mol), orthorhombic, *a* = 7.20942(19) Å, *b* = 11.7688(3) Å, *c* = 37.0363(10) Å, U = 3142.38(15) Å3, T = 115.85(10), space group P2_1_2_1_2_1_ (no. 19), Z = 4, μ (Cu Kα) = 2.908, 10984 reflections measured, 5938 unique (Rint = 0.0324), which were used in all calculations. The final R1 (I > 2σ (I), i.e., F_O_ > 4σ (F_o_)) was 0.0434, and the wR_2_ was 0.1161. The final wR (F^2^) was 0.1193 (all data). Flack parameter = −0.004(13). Detailed parameters are shown in Table 2. The crystallographic data were deposited at the Cambridge Crystallographic Data Center with the deposition number CDCC 2119390. Copies of the data can be obtained, free of charge, on application to CCDC, 12 Union Road, Cambridge CB2 1EZ, U.K. (fax +44(0)-1233-336033; email: deposit@ccdc.cam.ac.uk).

### 2.4. Anti-Tumor Activity against Seven Cancer Cell Lines

#### 2.4.1. Cytotoxic Activities

The cytotoxic activities of compounds **1** and **2** against a mouse melanoma cell line (B16F10), a human lung cancer cell line (A549), a human large cell lung cancer cell line (H460), a human breast cancer cell line (MCF–7), a human hepatocellular carcinoma cell line (HepG2), a human cervical carcinoma cell line (Hela), and a human glioma cell line (U87) were assayed using CCK–8 with cisplatin as the positive control. The results showed that compound **1** exhibited potent cytotoxic activities against HepG2 and B16F10, with 50% inhibitory concentration (IC_50_) values of 12.40 μM and 17.51 μM, respectively (Table 3). The IC_50_ values for the different tumor cells were as follows: HepG2 < B16F10 < MCF–7 < Hela < U87 < A549 < H460. However, compound **2** did not exhibit any cytotoxic activities on any of the seven cell lines. Comparative analysis of the structures of compounds **1** and **2** revealed that the acetal structure and double bond positions were the main differences between them, and these are presumed to account for the differences in cytotoxic activities. In other words, we believe that a C–30 acetal unit and a C–13–C–18 double bond are essential features for producing cytotoxic activities.

#### 2.4.2. Cell Morphology

To assess whether compound **1** could induce any morphological changes, HepG2 cells were incubated with different concentrations of compound **1**. The cells were then observed and photographed using a fluorescence microscope after being stained with 3,3′-dioctadecyloxacarbocyanine perchlorate (Dio) and Hoechst 33342 [12,13]. As shown in Figure 4A, compound **1** altered the cellular morphology considerably regardless of the dose. The untreated HepG2 cells (control group) exhibited a regular appearance, intensive growth, integral cell membrane, and clear nucleolus. After treatment with compound **1**, cell membrane and nuclear condensation can be observed, and the cells were eventually destroyed, showing elongated or irregular shapes, condensation of chromatin, and cell shrinkage, which revealed that compound **1** could induce apoptosis in vitro.

#### 2.4.3. Cell Apoptosis and Cell Cycle

In order to further explore the mechanism of compound **1** against tumors, HepG2 cells were incubated with different concentrations of compound **1** to observe its effect on cell apoptosis and cell cycle. The results showed that compound **1** could increase the cells in the G2/M phase from 17.53% to 22.05% (3.13 μM), 26.03% (6.25 μM), and 36.48% (12.5 μM) (Figure 4B,C), and the apoptosis rate from 14.31% to 24.00% (3.13 μM), 30.06% (6.25 μM), and 54.27% (12.5 μM) (Figure 4D,E) in a dose–dependent manner. Compared with the control group, compound **1** at concentrations of 6.25 μM and 12.5 μM significantly increased both the cells in the G2/M phase and the apoptosis rate (*p* < 0.01 or *p* < 0.001). Therefore, we speculated that compound **1** might inhibit cell proliferation by blocking the cell G2/M phase and promoting cell apoptosis.

### 2.5. Anti–Inflammatory Activity against NO Production

The anti-inflammatory assays of compounds **1** and **2** were evaluated in LPS–stimulated RAW264.7 macrophages with dexamethasone as the positive control. The results showed that compound **1** produced a cytotoxic effect on RAW264.7 cells. Therefore, the anti–inflammatory activity of compound **1** has not been tested. The results also showed that compound **2** did not affect cell viability on RAW264.7 cells at a concentration of 50 μM to 6.25 μM (Figure 5B). As shown in Figure 5C, the results showed that compound **2** exhibited good inhibitory activity against NO production in a dose-dependent manner. In addition, we used a microscope to observe the effect of compound **2** on the cell morphology of LPS–stimulated RAW264.7 cells in order to more accurately evaluate its anti-inflammatory properties. As can be seen in Figure 5A, the cells in the control group were plump and round, but the cells undergoing LPS stimulation became flat and irregular, with slender tentacles. However, the LPS–stimulated cell morphology returned to normal when compound **2** intervened. The above results demonstrate that compound **2** exhibits a potential anti-inflammatory activity by decreasing the production of NO in LPS–stimulated RAW264.7 cells.

## 3. Material and Methods

### 3.1. General Experimental Procedures

Silica gel (200–300 and 400–600 mesh, Qingdao Marine Chemical Co., Qingdao, China) and Sephadex LH–20 (Pharmacia Co., Stockholm, Sweden) were used for column chromatography (CC) and thin-layer chromatography (TLC). Analytical HPLC was performed on a Surveyor high performance liquid chromatography system (Thermo Fisher Co., Waltham, MA, USA) with an Alltech 2000ES evaporative light scattering detector (Alltech Co., Nicholasville, KY, USA). Preparative HPLC (Rp–C_18_: 21.2 × 150 mm, 5 μm, Innoval C_18_, Agela Technologies Co. Ltd., Tianjin, China) was carried out on a KNAUER apparatus with a refractive index detector (Knauer Co., Berlin, Germany). IR data were recorded on a Bruker Tensor–27 spectrometer (Bruker Co., Billerica, MA, USA). High–resolution electrospray ionization mass spectrometry (HR-ESI-MS) was performed on the Agilent G6230A TOF LC/MS (Agilent Technologies Co. Ltd., Santa Clara, CA, USA). Both 1D and 2D NMR spectra were recorded on a Bruker ECA–400 MHz spectrometer (Bruker Co., Billerica, MA, USA) with TMS as an internal standard, and chemical shifts are expressed in d (ppm). ECD spectra were recorded in methanol using a JASCO J–815 spectrophotometer (Jasco International Co., Ltd., Tokyo, Japan) at room temperature.

### 3.2. Plant Materials

The roots of *Ardisia lindleyana* D. Dietr were collected from Guangxi (104°26′–112°04′ E, 20°54′–26°24′ N at an altitude of about 300 m) in China, and identified by Prof. Bin Li, Beijing Institute of Radiation Medicine. A voucher specimen (Voucher # 2018–0808) was deposited in the specimen cabinet of the Beijing Institute of Radiation Medicine.

### 3.3. Isolations of Compounds

The root bark (60 kg) was extracted with 80% EtOH. Upon concentration with a rotary evaporator, the root bark yielded a dark, brownish black residue (11.8 kg). This residue was then dispersed in deionized water and partitioned successively with petroleum ether, EtOAc, and n–BuOH. The EtOAc extract (290 g) was subjected to a silica gel column using Dichloromethane–MeOH (200–300 mesh, 100:0, 100:1, 50:1, 25:1, 10:1, 5:1, 2:1, and 1:1, *v*/*v*) to produce 12 fractions (Fr.1–Fr.12). Fr.2 (8.5 g) was fractionated and subjected to a silica gel column (200–300 mesh, Petroleum ether–ethyl acetate, 100:0, 100:1, 50:1, 25:1, 10:1, 5:1, 2:1, and 1:1, *v*/*v*) to produce 15 fractions (Fr.2–a–Fr.2–o). Fr.2–d (1.5 g) was further separated by preparative HPLC using MeOH–H_2_O (85:15) to yield compound **1** (52.8 mg), and Fr.2–e (0.8 g) was further separated by preparative HPLC using MeOH–H_2_O (65:35, *v*/*v*) to yield compound **2** (35.6 mg).

### 3.4. Effect of Isolated Compounds on Cytotoxicity of Seven Cancer Cell Lines

Seven cancer cell lines, the B16F10, A549, H460, MCF–7, HepG2, Hela, and U87 cancer cell lines, were utilized in the cytotoxic activities assay. The MCF–7, HepG2, Hela, and U87 cell lines were cultured in Dulbecco’s modified Eagle’s medium (DMEM, Gibco, Grand Island, NY, USA), and The B16F10, A549, and H460 cell lines were cultured in RPMI 1640 medium (RPMI 1640, SIGMA, U.S.), in each case supplemented with 10% fetal calf serum (Gibco, Grand Island, NY, USA), 100 U/mL penicillin, and 100 μg/mL streptomycin in an atmosphere containing 5% CO_2_ at 37 °C. Cells were seeded in 96–well plates with a density of 5 × 10^4^ cells/mL and treated with different concentrations of the test compounds for 48 h. The cytotoxicity assay was performed using the Cell Counting Kit–8 (CCK–8, Applygen technologies Co., Ltd., Beijing, China) assay.

### 3.5. Effect of Compound 1 on Cell Morphology, Cell Cycle, and Apoptosis of HepG2 Cells

HepG2 cells were dispensed in a 12-well plate at a density of 5 × 10^4^ cells/mL, incubated at 37 °C for 24 h, and treated with different concentrations of compound **1** for 48 h.

#### 3.5.1. Cell Morphology Assay

After removing the supernatant, the cells were washed twice with 1.0 mL PBS and fixed with 4% Paraformaldehyde fix solution (Shanghai Biyuntian Biotechnology Co., Ltd., Shanghai, China) for 2 h. Fixed cells were then washed twice with 1.0 mL PBS and incubated with Dio (Shanghai Biyuntian Biotechnology Co., Ltd., Shanghai, China) at 37 °C in darkness. After 20 min of incubation, the cells were washed twice with 1.0 mL PBS. Then, the cells were stained with Hoechst 33342 staining solution (Shanghai Biyuntian Biotechnology Co., Ltd., Shanghai, China) for 10 min and washed twice with 1.0 mL PBS. The stained cells in PBS were then observed and photographed using a fluorescence microscope.

#### 3.5.2. Cell Apoptosis and Cell Cycle Assay

In the cell apoptosis assay, all the cells were collected, and 100,000 cells were taken and incubated with Annexin V–FITC and PI for 20 min at room temperature without light. In the cell cycle assay, all the cells were collected and fixed with precooled 70% ethanol for 12 h at 4 °C. The cells were then washed with PBS and incubated with PI for 30 min at 37 °C without light. All the cells were detected using a flow cytometer.

### 3.6. Effect of Isolated Compounds on NO Production

A RAW264.7 murine macrophage cell line (American Type Culture Collection, ATCC No. TIB-71) was purchased from the Cancer Institute and Hospital of the Chinese Academy of Medical Sciences. Cells were cultured in DMEM medium (Gibco, Grand Island, NY, USA) supplemented with 10% heat-inactivated fetal calf serum (Gibco, Grand Island, NY, USA), 100 U/mL penicillin, and 100 μg/mL streptomycin in an atmosphere containing 5% CO_2_ at 37 °C. The cells were cultured for 3–5 days to reach the logarithmic phase and then used for experiments. The cells were treated with tested compounds and a positive control (dexamethasone) at different concentrations and then stimulated with LPS (1 μg/mL) for 24 h.

#### 3.6.1. Cell viability Assay on RAW264.7 Cells

The cytotoxic effects of compounds **1** and **2** on RAW264.7 cells were evaluated using a CCK–8 assay. RAW264.7 cells were dispensed in a 96–well plate at a density of 1 × 10^5^ cell per well, incubated at 37 °C for 24 h, and treated with the tested agents for the indicated periods of time. Then, 10 µL of cell counting kit–8 solution was added to each well, followed by 1 h incubation. The absorbance at 450 nm of each well was measured with a Multiskan MK–3 microplate reader (Thermo Fisher Co., Waltham, MA, USA). Compound **2** showed no cytotoxic effect on RAW264.7 cells at different concentrations (6.25, 25, 50 µM). In addition, cell morphology was observed under an OLYMPUS CKX53 microscope (Olympus Co., Tokyo, Japan).

#### 3.6.2. Determination of the NO Content

NO production was determined using a nitric oxide assay Kit (Applygen technologies Co., Ltd., Beijing, China) based on the Griess reaction principle [14,15]. A quantity of 50 microliters of culture supernatant was transferred to 96–well plates and mixed with 50 microliters of Griess R1 reagent after incubation at room temperature for 5 min. The samples were then mixed with an equal volume Griess R2 reagent for another 5 min. The absorbance of each mixture at 540 nm was measured using a microplate reader [16].

#### 3.6.3. Cell Morphology Assay

Without any post-processing, the cells were observed and photographed using a microscope.

## 4. Conclusions

In this study, we identified two new compounds, including ardisiapunine B (**1**) and ardisiapunine C (**2**), which we isolated from *Ardisia lindleyana* D. Dietr. The structures of the two new compounds were elucidated using 1D and 2D NMR spectroscopic analyses, HR–ESI–MS, single–crystal X–ray diffraction, and ECD calculation. It was found that the two new compounds are unusual oleanane-type triterpenes, with compound **1** bearing an acetal unit and a C–13–C–18 double bond, and compound **2** bearing a C–28 aldehyde group and a C–18–C–19 double bond. Activity assay results showed that compound **1** exhibited a potent cytotoxicity against HepG2 and B16F10 cell lines, and that, furthermore, it might inhibit cell proliferation by blocking the cell G2/M phase and promoting cell apoptosis. These new findings provide a framework for the further exploration of compound **1**, which exhibits potential anti–tumor activity by inducing cell cycle arrest and apoptosis. Compound **2** had moderate inhibitory effects on the production of NO in LPS–stimulated RAW264.7 cells. Comparative analysis of the structures of compounds **1** and **2** revealed that the acetal structure and different double bond positions were the main differences between them, and these are presumed to be the main reasons for the extreme differences in their cytotoxicity and anti–inflammatory activities. From these new findings, two promising lead compounds were identified for the future development of potential anti-inflammatory or anti-tumor agents.

## Figures and Tables

**Figure 1 molecules-27-06641-f001:**
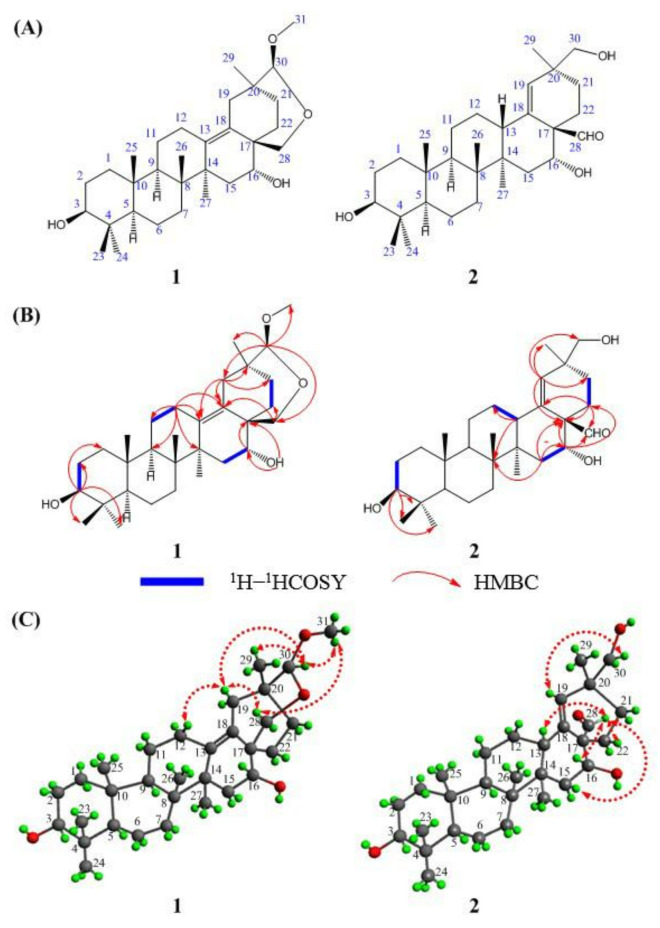
Chemical structures (**A**), key ^1^H–^1^HCOSY and HMBC correlations (**B**), and NOESY correlations (**C**) of compounds **1** and **2**.

**Figure 2 molecules-27-06641-f002:**
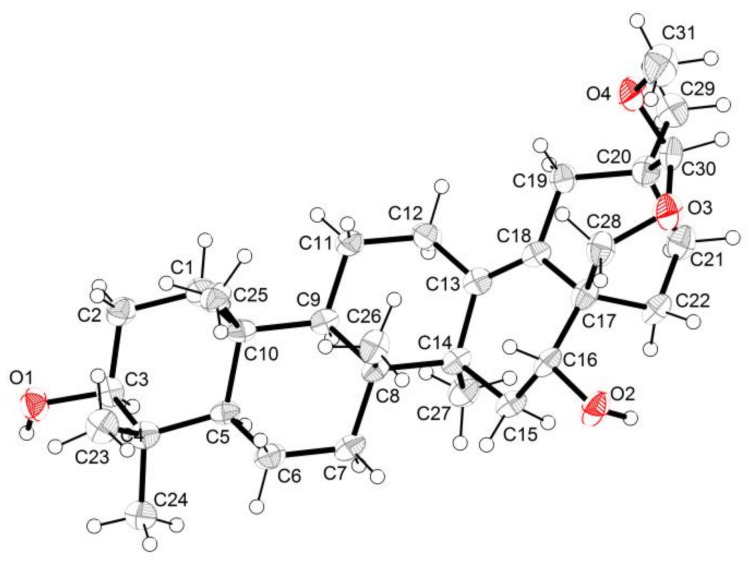
X-ray structure of compound **1**.

**Figure 3 molecules-27-06641-f003:**
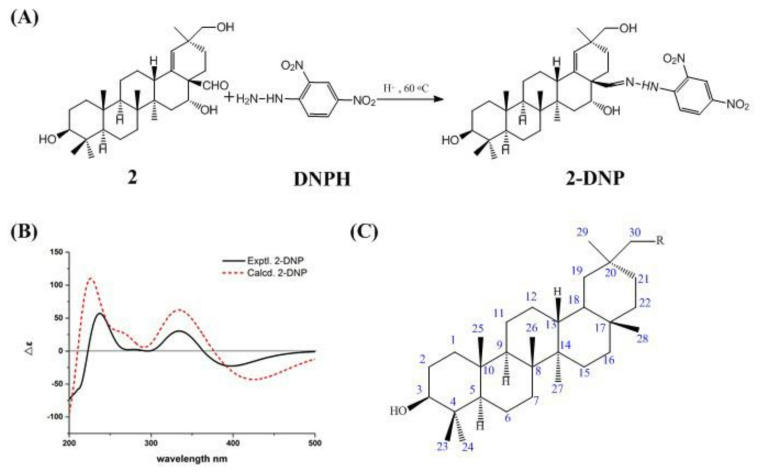
Mechanism of the reaction of DNPH with the aldehyde group of compound **2** (**A**). Experimental ECD spectra of compound **2–DNP** and calculated ECD spectra of compound **2–DNP** (**B**). Basic chemical structure of 3-hydroxyl oleanane–type pentacyclic triterpenes (**C**).

**Figure 4 molecules-27-06641-f004:**
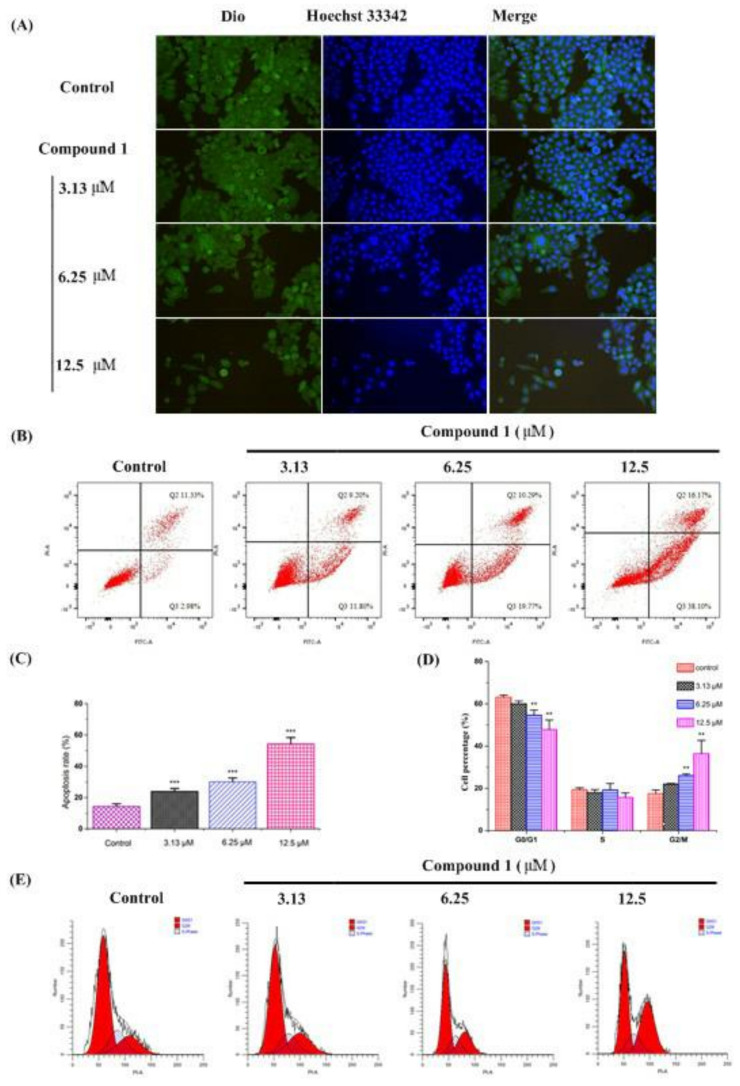
Effects of compound **1** on cell morphology, cell apoptosis, and cell cycle of HepG2 Cells in vitro. HepG2 cells were seeded in 12–well plates. After 24 h incubation, cells were incubated with compound **1** for another 48 h. Cell morphology assay (**A**): The cells were washed with PBS, fixed by paraformaldehyde, washed with PBS, stained with Dio, washed with PBS, stained with Hoechst 33342, and washed with PBS. The cells were then observed and photographed using a fluorescence microscope. Cell apoptosis (**B**,**C**) and cycle (**D**,**E**) assay: In the cell apoptosis assay, all the cells were collected, and 100,000 cells were taken and incubated with Annexin V–FITC and PI for 20 min at room temperature without light. In the cell cycle assay, all the cells were collected and fixed with precooled 70% ethanol for 12 h at 4 °C. The cells were then washed with PBS and incubated with PI for 30 min at 37 °C without light. Finally, all the cells were detected using a flow cytometer. Data are presented as mean ± SD, n = 3. One way ANOVA, ** *p* < 0.001, *** *p* < 0.001, compared with control group.

**Figure 5 molecules-27-06641-f005:**
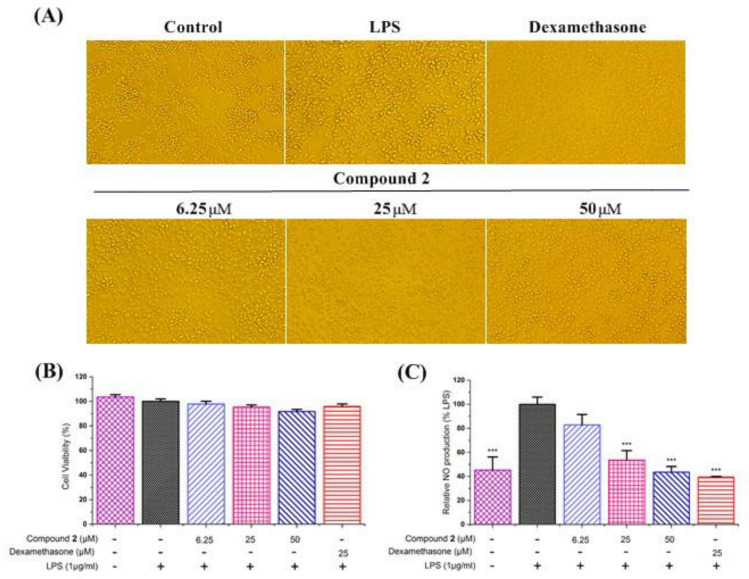
Effects of compound **2** on morphology (**A**), cell viability (**B**), and nitric oxide (NO) production (**C**) of LPS–stimulated RAW264.7 cells in vitro. After 24 h incubation, LPS–stimulated RAW264.7 cells were co-treated with LPS (1 μg/mL) and various concentrations of compound **2** for another 24 h. Dexamethasone was used as a positive control. In the cell morphology assay, the cells were observed and photographed using a microscope. Cell viability was determined using the CCK–8 method by measuring the absorbance with a microplate photometer at 450 nm. In (**B**,**C**), the abscissa groups from left to right are the control group (without LPS), model group, compound **1** at a concentration of 6.25 μM group, compound **1** at a concentration of 25 μM group, compound **1** at a concentration of 50 μM group and positive control group, respectively. In the NO production assay, the supernatants were collected for the measurement of NO production using the Griess reagent. Data are presented as mean ± SD, n = 3. One way ANOVA, *** *p* < 0.001, compared with LPS group.

**Table 1 molecules-27-06641-t001:** NMR data of compounds **1**, **2**, and **2-DNP** in pyridine (400 MHz for ^1^H; 100 MHz for ^13^C).

NO	1		2		2-DNP	
	*δ*_C_, Mult.	*δ*_H_ (*J* in Hz)	*δ*_C_, Mult.	*δ*_H_ (*J* in Hz)	*δ*_C_, Mult.	*δ*_H_ (*J* in Hz)
1	40.8, CH_2_	overlapped	39.1, CH_2_	overlapped	39.3, CH_2_	overlapped
2	29.6, CH_2_	overlapped	28.1, CH_2_	overlapped	28.1, CH_2_	overlapped
3	79.3, CH	3.47, t (7.5)	77.8, CH	3.42, m	77.8, CH	3.44, t (7.5)
4	40.9, C		39.3, C		39.3, C	
5	57.4, CH	overlapped	55.8, CH	overlapped	55.7, CH	overlapped
6	20.2, CH_2_	overlapped	18.4, CH_2_	overlapped	18.4, CH_2_	overlapped
7	36.1, CH_2_	overlapped	34.8, CH_2_	overlapped	34.7, CH_2_	overlapped
8	43.2, C		42.4, C		42.4, C	
9	52.8, CH	overlapped	50.6, CH	overlapped	50.7, CH	overlapped
10	38.9, C		37.2, C		37.2, C	
11	23.5, CH_2_	overlapped	20.9, CH_2_	overlapped	21.1, CH_2_	overlapped
12	27.1, CH_2_	2.57, d (13.7), Ha	26.1, CH_2_	overlapped, Ha	26.2, CH_2_	overlapped, Ha
		overlapped, Hb		overlapped, Hb		overlapped, Hb
13	134.1, C		40.7, CH	2.67, d (9.4)	40.3, CH	2.71, d (9.6)
14	45.7, C		40.8, C		40.7, C	
15	38.1, CH_2_	overlapped	39.6, CH_2_	overlapped	39.5, CH_2_	overlapped
16	72.1, CH	overlapped	75.5, CH	4.12, dd (4.6, 11.9)	75.8, CH	4.12, d (11.4)
17	45.8, C		57.0, C		50.0, C	
18	131.7, C		137.4, C		139.5, C	
19	36.9, CH_2_	3.16, d (16.3), Ha	132.1, CH	5.65, s	131.5, CH	5.70, s
		1.97, d (16.4), Hb				
20	40.7, C		37.9, C		38.2, C	
21	32.6, CH_2_	overlapped	28.5, CH_2_	overlapped	30.0, CH_2_	overlapped
22	25.9, CH_2_	2.24, m, Ha	26.9, CH_2_	2.88, m, Ha	27.9, CH_2_	2.80, m, Ha
		overlapped, Hb		overlapped, Hb		overlapped, Hb
23	30.0, CH_3_	1.20, s	28.4, CH_3_	1.19, s	28.4, CH_3_	1.19, s
24	17.7, CH_3_	1.02, s	16.1, CH_3_	1.07, s	16.1, CH_3_	1.01, s
25	18.3, CH_3_	0.88, s	16.7, CH_3_	0.85, s	16.7, CH_3_	0.83, s
26	20.2, CH_3_	1.00, s	16.1, CH_3_	1.01, s	16.1, CH_3_	1.01, s
27	27.4, CH_3_	1.40, s	16.0, CH_3_	0.95, s	16.4, CH_3_	1.00, s
28	72.2, CH_2_	overlapped, Ha	205.9, CH	10.39, s	157.4, CH	8.65, s
		3.82, t (4.0), Hb				
29	29.9, CH_3_	0.98, s	23.9, CH_3_	1.24, s	23.9, CH_3_	1.30, s
30	109.7, CH	4.40, s	71.5, CH_2_	3.68, m	72.1, CH_2_	3.68, dd (10.2, 25.2)
31	56.2, CH_3_	3.40, s				
1′					145.6, C	
2′					129.0, C	
3′					122.7 (overlapped), CH	9.04, d (2.5)
4′					137.2, C	
5′					129.6, CH	8.19, dd (2.5, 9.6)
6′					116.7, CH	8.12, d (9.6)

**Table 2 molecules-27-06641-t002:** Crystal data and structure refinement of compound **1**.

Parameter	Data
Empirical formula	C_32_H_51_C_l3_O_4_
Formula weight	606.08
Temperature / K	115.85(10)
Crystal system	orthorhombic
Space group	P2_1_2_1_2_1_
a / Å, b / Å, c / Å	7.20942(19), 11.7688(3), 37.0363(10)
α/°, β/°, γ/°	90.00, 90.00, 90.00
Volume / Å^3^	3142.38(15)
Z	4
ρcalc / mg mm^−3^	1.281
μ / mm^−1^	2.908
F(000)	1304
Crystal size / mm^3^	0.34 × 0.33 × 0.13
2Θ range for data collection	7.88 to 142.04°
Index ranges	−7 ≤ h ≤ 8, −14 ≤ k ≤ 10, −44 ≤ l ≤ 32
Reflections collected	10984
Independent reflections	5938[R(int) = 0.0324 (inf-0.9Å)]
Data/restraints/parameters	5938/3/37^1^
Goodness-of-fit on F^2^	1.050
Final R indexes (I > 2σ (I), i.e., F_o_ > 4σ (F_o_))	R_1_ = 0.0434, wR_2_ = 0.1161
Final R indexes (all data)	R_1_ = 0.0461, wR_2_ = 0.1193
Largest diff. peak/hole / e Å^−3^	0.497/−0.373
Flack Parameters	−0.004(13)
Completeness	0.9982

**Table 3 molecules-27-06641-t003:** Cytotoxic activity of compounds **1** and **2** against seven tumour cell lines in vitro.

Compound	IC_50_ (μM)						
	B16F10 ^a^	A549 ^b^	H460 ^c^	MCF–7 ^d^	HepG2 ^e^	Hela ^f^	U87 ^g^
**1**	17.51 ± 3.55	29.86 ± 4.26	33.09 ± 0.58	21.73 ± 2.71	12.40 ± 3.76	24.96 ± 6.88	25.50 ± 3.45
**2**	>100	>100	>100	>100	>100	>100	>100
cisplatin	9.07 ± 0.66	8.45 ± 0.74	9.63 ± 0.49	8.23 ± 0.43	9.89 ± 0.61	8.26 ± 0.51	8.89 ± 0.63

Note: Data are presented as means ± SD, n = 3; ND, no detection; ^a^ mouse melanoma cell line (B16F10); ^b^ human lung cancer cell line (A549); ^c^ human large cell lung cancer cell line (H460); ^d^ human breast cancer cell line (MCF–7); ^e^ human hepatocellular carcinoma cell line (HepG2); ^f^ human cervical carcinoma cell line (HeLa); ^g^ human glioma cell line (U87).

## Data Availability

The data presented in this study are available in the Appendix A.

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
