# Peer review of "Absolute Configuration and Biological Evaluation of Novel Triterpenes as Possible Anti-Inflammatory or Anti-Tumor Agents"

_molecules, 2022, doi:10.3390/molecules27196641_

Round 1

Reviewer 1 Report

The authors carried out an interesting study at the intersection of organic and medicinal chemistry. New compounds belonging to triterpenes and of particular interest as potential biologically active substances for the development of modern anticancer drugs based on and using modern methods of flow cytometry have been discovered and characterized.

I think that this article will attract special attention among researchers in the field of isolation and targeted modification of triterpenoids and related compounds, as well as specialists in the field of organic and medicinal chemistry, as well as pharmacology. The article can be accepted for publication after the following comments have been eliminated and the reviewer's questions have been answered:

- It is known that a fairly large number of representatives of triterpenes are active in relation to one of the key enzymes of the cell cycle - topoisomerase [D'yakonov VA, Dzhemilev LU, Dzhemilev UM Advances in the Chemistry of Natural and Semisynthetic Topoisomerase I / II Inhibitors. Studies Nat Prod Chem (2017) 54:21-86. https://doi.org/10.1016/B978-0-444-63929-5.00002-4], I believe that this aspect can also be discussed in the introductory part of the article with the inclusion of this reference.

- The authors studied cytotoxicity on a wide range of tumor cell lines, while an important aspect is the selectivity index. How do your compounds affect normal or conditionally normal cell lines?

Author Response

Comment 1: It is known that a fairly large number of representatives of triterpenes are active in relation to one of the key enzymes of the cell cycle - topoisomerase [D'yakonov VA, Dzhemilev LU, Dzhemilev UM Advances in the Chemistry of Natural and Semisynthetic Topoisomerase I / II Inhibitors. Studies Nat Prod Chem (2017) 54:21-86. https://doi.org/10.1016/B978-0-444-63929-5.00002-4], I believe that this aspect can also be discussed in the introductory part of the article with the inclusion of this reference.

Response 1: Thank you for your valuable comments, we have made correction according to the comment.

Comment 2: The authors studied cytotoxicity on a wide range of tumor cell lines, while an important aspect is the selectivity index. How do your compounds affect normal or conditionally normal cell lines?

Response 2: The two compounds we isolated are new compounds, and their biological activities were unknown. Therefore, we selected as many tumor cell lines as possible to screen for cytotoxicity in order to obtain its selectivity for tumor cells, and these tumor cell lines are common and representative. Furthermore, we selected HUVEC (human umbilical vein endothelial cell) cells to evaluate the effect of these two compounds on normal cells, the results showed that compound 1 inhibited the growth of the normal cells at a concentration of 50 μM, but not at concentrations below 25 μM, and compound 2 did not inhibited the growth of both normal cells at a concentration of 100 μM.

Reviewer 2 Report

The manuscript entitled “Absolute configuration and biological evaluation of novel triterpenes as possible anti-inflammatory or anti-tumor agents” describes the isolation and characterization of two structures with their biological evaluation. The manuscript may be of general interest to the researchers of this field, but the manuscript lacks some information that the author should consider and incorporate in the present form of the manuscript. Here are a few concerns that need to be addressed in the present form of the manuscript.

Some comments and corrections for authors:

1.     Overally, the manuscript has a lot of punctuation and grammatical errors and needs to be corrected (i.e., there must be comma before and in all mns when mention about over two parameters). Please run throughout the mns.

2.     The introduction section must be expanded by inserting the importance of these structures, obtained methods etc.

3.     The crystallographic and structure refinement data for both structures should be summarized in a Table.

4.     “1H-NMR” and “13C-NMR” must be corrected as “1H NMR” and “13C NMR”. Please check the mns carefully.

5.     I would be more than happy if the authors place correlations in the abstract and conclusion since they are very important and exciting.

6.     Prior to biological evaluation, the purity of all molecules should be checked by HPLC.

Author Response

Comment 1: Overally, the manuscript has a lot of punctuation and grammatical errors and needs to be corrected (i.e., there must be comma before and in all mns when mention about over two parameters). Please run throughout the mns.

Response 1: Thank you for your valuable comments, we have made correction and marked in red according to the comment.

Comment 2:  The introduction section must be expanded by inserting the importance of these structures, obtained methods etc.

Response 2: Thank you for your valuable comments, we have made correction and marked in red as follow: The structure of new compounds 12 are novel as unusual oleanane-type triterpenes, with compound 1 bearing an acetal unit and a C-13—C-18 double bond, and compound 2 bearing a C-28 aldehyde group and a C-18—C-19 double bond. The characterization of 12 enriched not only the diversity of chemical constituents of Ardisia lindleyana D. Dietr, but also the diversity of oleanane-type triterpene.

Comment 3: The crystallographic and structure refinement data for both structures should be summarized in a Table.

Response 3: according to your comment, we have made correction and marked in red. One point to be noted is that there are the crystallographic and structure refinement data for only compound 1, because the absolute configuration of compound 1 was determined by single-crystal diffraction techniques, while compound 2 did not successfully cultivate the single crystals, whose absolute configuration was determined by ECD calculations.

Comment 4: “1H-NMR” and “13C-NMR” must be corrected as “1H NMR” and “13C NMR”. Please check the mns carefully.

Response 4: Thank you for your valuable comments, we have made correction according to the comment.

Comment 5: I would be more than happy if the authors place correlations in the abstract and conclusion since they are very important and exciting.

Response 5: Thank you for your constructive comments, we have placed correlations in the abstract and conclusion and marked in red.

Comment 6: Prior to biological evaluation, the purity of all molecules should be checked by HPLC.

Response 6: according to your valuable comment, we add these descriptions “The purity of compound 1 was more than 95% by HPLC analysis” and “The purity of compound 2 was more than 95% by HPLC analysis.” that marked with red in the text, and the HPLC charts of compounds 1-2 are as shown in Figure S9, Figure S10, Figure S20 and Figure S21 in the Supplementary material.

Reviewer 3 Report

The article is about the isolation of two new triterpenoid compounds from Ardisia lindleyana, establishing of their exact structure including the configuration at their chiral atoms.

In my opinion, the article is well written and the experiments are well designed to fulfill the target. It seems to me, that the structures were determined in a correct way by several independent methods including 2D NMR spectra, X-ray for the compound 1 and by ECD of the derivatized compound 2. Pictures of the NMR in the supporting material are clean with minimum of impurities. In addition, the cytotoxicity of the compounds was measured in several cell lines but the activity is just moderate.

Overall, this is a nice article that may be interesting for the readers of the journal although the biological activities are just moderate. The main contribution to the general knowledge lays in the isolation and elucidation of the new structures. To my best knowledge, the spectral data correspond to the proposed structures. Biology is well done, I agree with the block at the G2/M phase and the apoptotic pathway and with the fact that compound 2 reduces the production of NO moderately. 

Author Response

Dear reviewer:

Thank you very much for your valuable comments. Thank you.